

# Algorithm design of a combinatorial mathematical model for computer random signals

Qinghua Yao[1] and Benhua Qiu[2]

[1] Xuchang Vocational College of Ceramic, Xuchang, Henan, China
[2] Department of Basic Courses, Zhengzhou University of Science and Technology, Zhengzhou, Henan, China

## ABSTRACT

To improve the processing effect of computer random signals, the manuscript employs the intelligent signal recognition algorithm to design a combinatorial mathematical model for computer random signals, and studies the parameter estimation of conventional frequency hopping signal (FHS) based on optimizing kernel function (KF). First, the mathematical form and graphical representation of the ambiguity function of the conventional FHS are explored. Furthermore, a new KF is presented according to its fuzzy function (FF) and the parameters of conventional FHSs are estimated according to the time-frequency distribution corresponding to the KF. Then, simulation experiments are carried out in different types of interference noise environments. The proposed combinatorial mathematical model for computer random signals shows a practical impact, and can effectively improve the effect of random signal combination.

# INTRODUCTION

When the passive intelligent surveillance radar system is under consideration, the main purpose of the receiver is to analyze the modulation type of the radiated source signal, and correctly distinguish the linear frequency of the modulation signal as much as possible by employing other functional forms such as the cubic phase signal, or other high-order pulse modulation signals (*Edla et al., 2018*). In the communication system, the delay of the transmitted signal will change with the relative position of the transmitter and the receiver, so that the instantaneous phase during the transmission and reception process changes with the constant change of the distance (*Олйник & Лукн, 2020*). So, the received signal can be constructed by the polynomial phase signal. In addition, in high dynamic communication systems such as special communication and anti-jamming communication, secondary frequency modulation, which is a pseudo code phase modulation, generates composite signals whose information code is utilized to achieve low interception probability and better anti-jamming performance (*Mohdiwale et al., 2020*). The estimation of time-varying fading channels also employs chirps. In biomedical ultrasound systems, chirp signals and their pulse compression techniques are employed to improve the signal-to-noise ratio to

Corresponding author
Qinghua Yao,
yaoqinghua-2008@163.com

achieve the desired imaging depth while maintaining effective resolution (*Savchenko, 2018*). At the same time, the FM signal is also selected for seismic survey and digital watermark processing.

There are various methods for estimating the parameters of the third-order phase signal, and the parameter estimation can also be realized by employing the difference operation. The estimations of three parameters can be obtained after performing phase difference operation on the third-order phase signal (*Geng et al., 2022*). This method is simple and requires less computation. However, when the signal is disturbed by noise, the difference operation will be very sensitive to the noise, so the sliding window method (*Roy et al., 2022*) was employed to average the processing, thereby reducing the mean square error of parameter estimation.

In *Yuqing et al. (2019)*, the researchers first proposed the method of polynomial phase transformation to realize the parameter estimation of the polynomial phase signal. When the signal amplitude is not constant or the amplitude changes rapidly, the estimation performance of the algorithm will be affected. The feature of the polynomial phase transformation estimates the high-order phase signal parameters successively by utilizing the low-order phase parameters (*Geran Malek, Mansoori & Omranpour, 2021*). Due to the implementation of the reduced-order form, the errors of the high-order parameters will be transferred to the low-order parameters in turn. At the same time, the range of the parameter estimations is narrower when this method is implemented.

In *Geran Malek, Mansoori & Omranpour (2021)*, two solution methods were proposed to expand the range of the parameter estimations. The aliasing algorithm, non-uniform sampling method, and guided selection algorithm can all realize parameter estimation of the single-component signal. Signal parameter estimation can also be achieved by time-frequency analysis methods. The Wigner distribution method proposed in *Williamson, Fazli & Lee (2018)* administers the time-frequency analysis method to track the time-frequency curve of the single-component third-order phase signal to obtain frequency information. In *Marulanda AH & Vega (2020)*, a cubic phase function method was proposed to estimate the third-order phase signal. The algorithm only needs to realize the extreme value search when one-dimensional conditions are used, and the asymptotic statistical method is employed to effectively improve the accuracy of parameter estimations. When compared with the fourth-order nonlinear transformation in the polynomial phase signal parameter estimation, the second-order nonlinear transformation implemented in the third-order phase transformation improves the noise suppression performance of the algorithm. The mean square error of the parameter estimations is approximated by the Cramer-Rao bound line, which can effectively realize the parameter estimations. A random cubic phase function transformation method based on the third-order phase signal was proposed in *Panchenko & Pechenyuk (2019)*.

The algorithm can achieve a lower threshold and the mean square error of the parameters can reach the Cramer-Rao bound line. The research regarding the parameter estimation of single-component polynomial phase signals is mainly run to improve the estimation performance when a low signal-to-noise ratio exists. The third-order phase function has already demonstrated a good ability to resist noise interference, but to better obtain

weak signals in practical engineering, *Salih, Tawfeeq & Khaleel (2019)* proposed a cubic phase function extension method, which improved the estimation performance when SNR was low by implementing the correlation integration in the two-dimensional CPF transform space. Furthermore, data analysis showed that the computational complexity of the algorithm is moderate, much less than the maximum likelihood estimation. To reduce the mean square error of single-component phase modulation parameter estimation, a fourth-order phase function expansion method based on the fourth-order nonlinear transformation was proposed in *Kumar & Chang (2020)* with the advantage of infinite approximation of lines. *Gu et al. (2021)* applied the algorithm to estimate the third-order phase signal, but the estimation performance of this method is sensitive to noise interference, and it is only suitable for single-component signals, so the algorithm needs to be able to suppress noise and deal with multi-component signals. Some higher-order phase function parameter estimation methods are also applicable to third-order phase signals. The standard high-order phase function was proposed in *Akgün (2022)*. Due to multiple times of data introduction and searching for the minimum mean square error, this method has better performance advantages than high-order FFs and polynomial Gana distribution methods. But it can be only applied to a single-component signal.

Parameter estimation of multi-component signals was developed after the research on single-component signals was relatively mature. *Khan et al. (2021)* presented the Cramer bound for the parameter estimations of multi-component signals, and the correctness of the bound is confirmed by experiments. Early research on parameter estimation of multi-component and multi-phase signals mainly focused on the field of time-frequency analysis. Time-frequency analysis is mainly implemented to analyze the time-frequency variation characteristics of the signal, and it is an important parameter estimation method. The Gener distribution combined random transformation method. *Gupta, Chopda & Pachori (2019)* transformed the time-frequency domain in the results of the Gener distribution by utilizing a two-dimensional structural domain, thereby suppressing interference and obtaining parameter values. However, this method needs to resolve the issue of computational complexity due to the two-dimensional transformation. *Balaji et al. (2020)* proposed an extended method of nonlinear instantaneous frequency least squares approximation based on multi-component signals, but the disadvantage is that the algorithm is computationally inefficient and the optimal parameter selection for parameter estimation is unknown.

*Fira, Costin & Gora (2021)* employed the cubic phase function to correct the high-order ambiguity function, and the cubic phase function realized the cubic term function of the final stage, thereby reducing the signal-to-noise ratio threshold and the mean square error of parameter estimation. Due to its limitations, even if it is corrected by the cubic phase function, the amount of calculation in the entire parameter estimation process is still very large. The cubic phase function can realize the parameter estimation of the single-component signal, but for the multi-component signal, the estimation process is disturbed by interleaving terms, spurious peaks, *etc*. Therefore, if you want to use the cubic phase function to estimate the parameters of the multi-component signal, its anti-interference ability needs to be improved. *Tanaka, Ortega & Cheung (2020)* took advantage of the parameter estimation performance of the cubic phase function method when a

low signal-to-noise ratio exists and has helped make a detailed analysis of the interleaved term interference and pseudo-peak interference under multi-component conditions. By incorporating the idea of multiplication, the improved algorithm has various parameters of the multi-component chirp signal that can be effectively estimated, thereby improving the estimation performance of the chirp signal. More up-to-date research can be found in *Zhang (2020)*, *Martinez-Herrera et al. (2023)*.

The manuscript implements the intelligent signal recognition algorithm to design a combinatorial mathematical model for computer random signals to improve the stability of the signal transmission system in complex environments (*Edla et al., 2018*).

The outline of the article is as follows: 'The Algorithm of Random Signal Combination' presents the algorithm of random signal combination and the proposed method. The experiments and their results are given in 'Experimental Research'. The research is concluded in 'Conclusion'.

## THE ALGORITHM OF RANDOM SIGNAL COMBINATION

### Definitions and properties of FFs

If an inverse Fourier transform (IFT) is performed on the instantaneous correlation function concerning time t, another two-dimensional function, namely, an FF can be obtained. The definition of the FF is given in Eq. (1).

$$A(\tau,v) = \int_{-\infty}^{\infty} x\left(t + \frac{\tau}{2}\right) x^*\left(t - \frac{\tau}{2}\right) e^{jvt} dt \tag{1}$$

Therefore, a non-stationary signal can have two bilinear representations, both of which are the two-dimensional FT of the signal, and Eq. (2) presents two-dimensional interchanges.

$$A(\tau,v) = \int_{-\infty}^{\infty} \int_{-\infty}^{\infty} WVD(t,f) e^{j2\pi(tv+\tau f)} dt df \tag{2}$$

For a two-component signal, denoted by $x(t) = x_1(t) + x_2(t)$, the Wigner-Ville distribution of this signal and the positional relationship between the cross term and the self-term of the ambiguity function are shown in Fig. 1. Among them, the label of the ellipse is the cross term (coherent term), and the label of the rectangle is the self-term.

Figure 1 WVD of two-component signal and the positional relationship of self-term and cross-term in FF

The FF of the signal has the following properties:

(1) The time-shift invariance is denoted by Eq. (3).

$$A_{x(t-t_0)}(\tau,v) = e^{jvt_0} A_{x(t)}(\tau,v) \tag{3}$$

(2) The frequency shift invariance is represented by Eq. (4).

$$A_{x(t)e^{j\omega_0 t}}(\tau,v) = e^{j\omega_0 t} A_{x(t)}(\tau,v) \tag{4}$$

(3) The maximum value of the FF is always at the origin of the $(\tau,v)$ plane, and the maximum value is the energy of the signal characterized by Eq. (5).

$$maxA_x(\tau,v) = A_x(0,0) = \frac{1}{2\pi} E_x \tag{5}$$

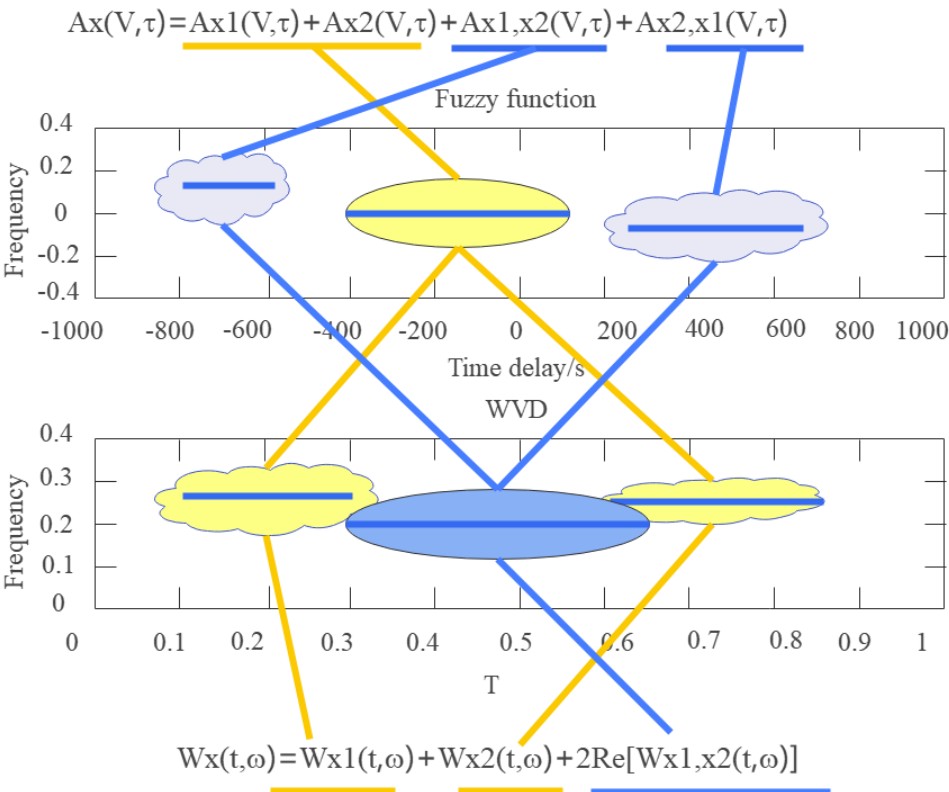

**Figure 1   WVD of two-component signal and the positional relationship of self-term and cross-term in FF.**

(4) The self-term of the FF is always distributed near the origin of the fuzzy domain (FD), while the coherent term of the FF is always distributed far from the origin. The greater the time or frequency difference between the two components, the further away the coherent term they form is from the origin of the FD.

### FD filtering and KF theory

The self-term of the FF of the signal is always concentrated in the center of the origin of the FD, and the coherent term of the FF is always far away from the center of the FD, so it can be filtered out by a two-dimensional low-pass filter.

Two-dimensional filtering is performed on the FF of the signal, and then a two-dimensional FT is performed on the $\tau$ and v variables to obtain the bilinear time-frequency distribution of the signal, also known as bilinear or Cohen-like time-frequency analysis. The simulation results show that the better the suppression of the cross-term, the worse the time-frequency resolution of the signal. The two-dimensional filter function in the FD is called a KF, and different KFs determine different time-frequency distributions.

Therefore, a method to realize the bilinear time-frequency distribution is the FD filtering method, which first calculates the instantaneous correlation function $r_z(t, \tau)$ of the signal

$z(t)$ given in Eq. (6).

$$r_z(t,\tau) = z\left(t+\frac{\tau}{2}\right)z^*\left(t-\frac{\tau}{2}\right) \tag{6}$$

Then, the IFT of the instantaneous correlation function about the variable t is calculated, and the FF is obtained and characterized by Eq. (7).

$$A_z(\tau,v) = \int_{-\infty}^{\infty} r_z(t,\tau)e^{jvt}dt \tag{7}$$

The characteristic function $M(\tau,v)$ isobtained by multiplying the FF and the KF defined in Eq. (8).

$$M_z(\tau,v) = \varphi(\tau,v)A_z(\tau,v) \tag{8}$$

The bilinear time-frequency distribution can be obtained by performing the two-dimensional FT of the product, that is, the characteristic function on the variable $\tau,v$ defined by Eq. (9).

$$C_z(t,\omega) = \iint_{-\infty}^{\infty} A_z(\tau,v)\varphi(\tau,v)e^{-j(vt+\omega\tau)}d\tau\,dv \tag{9}$$

The flow chart of the above FD filtering algorithm is shown in Fig. 2.

If the KF in Eq. (8) selects the exponential kernel denoted by Eq. (10).

$$\varphi(\tau,v) = e^{-[\alpha(\tau v)^2]}. \tag{10}$$

Using the MATLAB 2017 version with the simulation toolbox where $\alpha = 0.5$, the Choi-Williams distribution (CWD) of the signal will be obtained, as shown in Fig. 3.

In the simulation, the observation duration is assigned to 10 ms, the hopping period is set to 2 ms, the number of hops is assigned to 5, and the hopping frequency is randomly selected ranging between 5 kHz through 50 kHz.

The MATLAB 2017 implementation of the instantaneous correlation function, FF, and Choi-Williams distribution after two-dimensional filtering are shown in Fig. 4.

Through the above calculations and simulations, the FD filtering method can more intuitively reflect the relationship between the signal in the time-frequency domain and the FD.

## FF of the conventional FHS

Substituting Eq. (8) into Eq. (1), the FD representation of the conventional FHS is obtained.

$$A_x(\tau,v) = \int_{-\infty}^{\infty} x\left(t+\frac{\tau}{2}\right)x^*\left(t-\frac{\tau}{2}\right)e^{j2\pi tv}dt$$
$$= \int_{-\infty}^{\infty}\sum_{n=0}^{N-1} rect\left(t+\frac{\tau}{2}-nT_h-\alpha T_h\right)e^{2\pi f_n\left(t+\frac{\tau}{2}-nT_h-\alpha T_h\right)} \tag{11}$$
$$\sum_{m=0}^{N-1} rect\left(t+\frac{\tau}{2}-nT_h-\alpha T_h\right)e^{2\pi f_n\left(t+\frac{\tau}{2}-nT_h-\alpha T_h\right)}$$

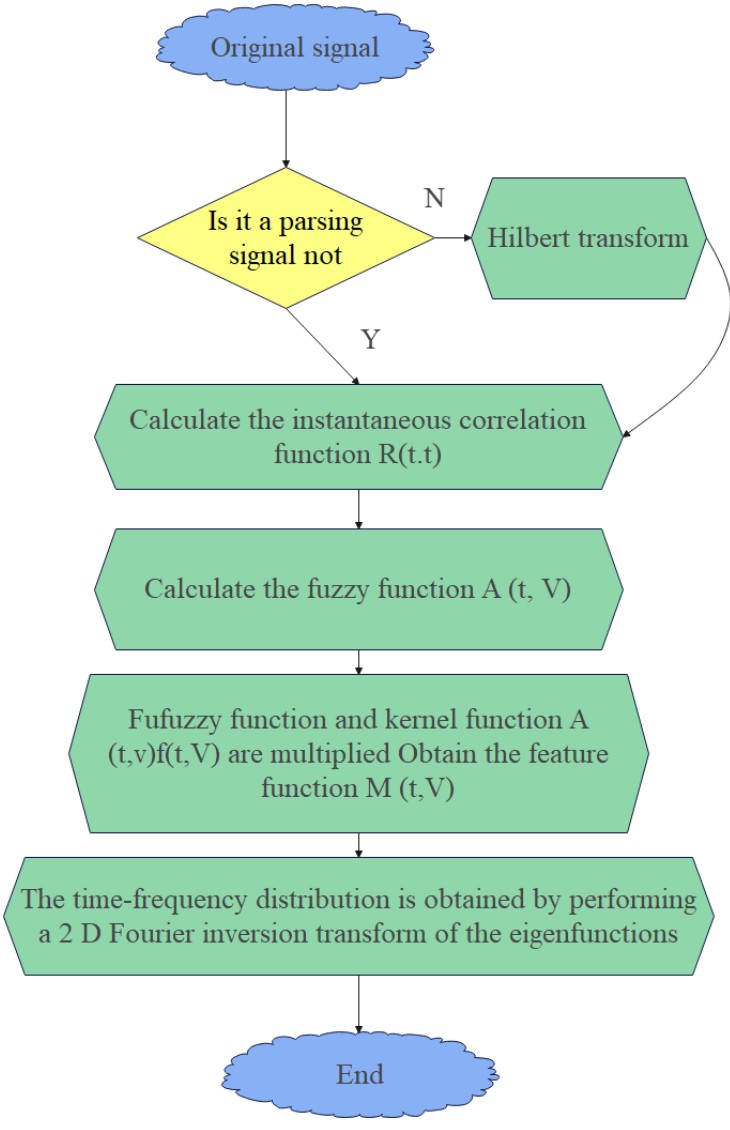

**Figure 2** Flowchart of the FD filtering algorithm.

$t + \frac{\tau}{2} - nT_h - \alpha T_h = t'$ is set to obtain:

$$A_x(\tau, v) = \int_{-\infty}^{\infty} \sum_{n=0}^{N-1} rect(t') e^{j2\pi f_n(t)} \sum_{m=0}^{N-1} rect(t' - \tau - mT_h + nT_h)$$

$$e^{j2\pi f_m(t' - \tau T_h + nT_h)} e^{j2\pi v(t' - \frac{v}{2} + nT_h + \alpha T_h)} dt$$

$$= \sum_{n=0}^{N-1} \sum_{m=0}^{N-1} e^{j2\pi f_m(\tau - nT_h + mT_h)} e^{j2\pi v(nT_h + \alpha T_h - \frac{\tau}{2})}.$$

$$\int_{-\infty}^{\infty} rect(t) rect(t - \tau - mT_h + nT_h) e^{j2\pi t(v - f_m + f_n)} dt$$

$$= \sum_{n=0}^{N-1} \sum_{m=0}^{N-1} e^{j2\pi f_m(\tau - nT_h + mT_h)} e^{j2\pi v(nT_h + \alpha T_h - \frac{\tau}{2})} * A_{ect}(\tau - nT_h + mT_h, v - f_m + f_n)$$

(12)

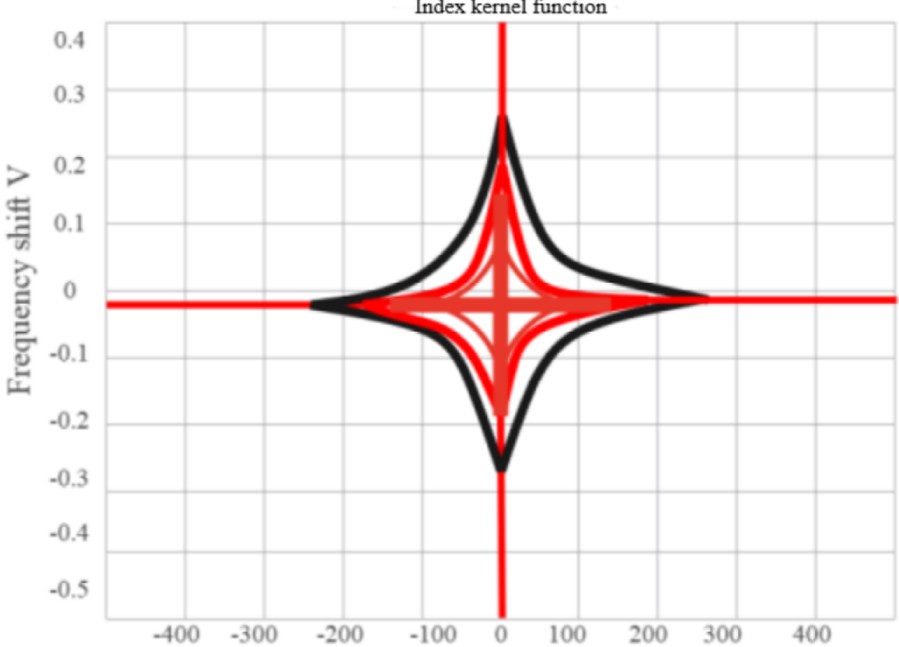

**Figure 3 Contour map of exponential kernel FD.**

Among them, $A_{\text{rect}}(.)$ denotes the FF of the square wave pulse signal $rect\left(\frac{t}{T_h}\right) = \begin{cases} 1, |t| < \frac{T_h}{2} \\ 0, others \end{cases}$, and the width is denoted by $T_H$, namely,

$$A_{rect}(\tau, v) = \begin{cases} e^{j2\pi v T_h} \dfrac{sin\pi v (T_h - |\tau|)}{\pi v}, |\tau| < T_h \\ 0, |\tau| > T_h \end{cases}. \tag{13}$$

The self-term of the FF of each component has the same delay direction width. Figure 5 presents a contour plot of the magnitude of the ambiguity function for a conventional FHS. Among them, the darker and thicker color in the center of the FD is the self-term of the FF, which needs to be retained, while the others are coherent terms, corresponding to the cross terms of the time-frequency distribution, which should be eliminated.

When located near the origin of the FD it is the self-term of the FF, and the length of the self-term delay of the FF in the conventional FHS is proportional to the period of the conventional FHS.

## KF design criteria and methods

Through the previous analysis, the designed KF should preferably completely suppress the coherent term in the FD, which needs to be appropriately truncated in the frequency-shift direction.

According to the design criteria of the FD-KF, the designed KF is defined in Eq. (14).

$$\varphi(\tau, \text{v}) = \frac{sin(\alpha\pi\text{v}(\beta - |\tau|))}{\alpha\beta\pi\text{v}}, \alpha \neq 0, \beta \neq 0 \tag{14}$$

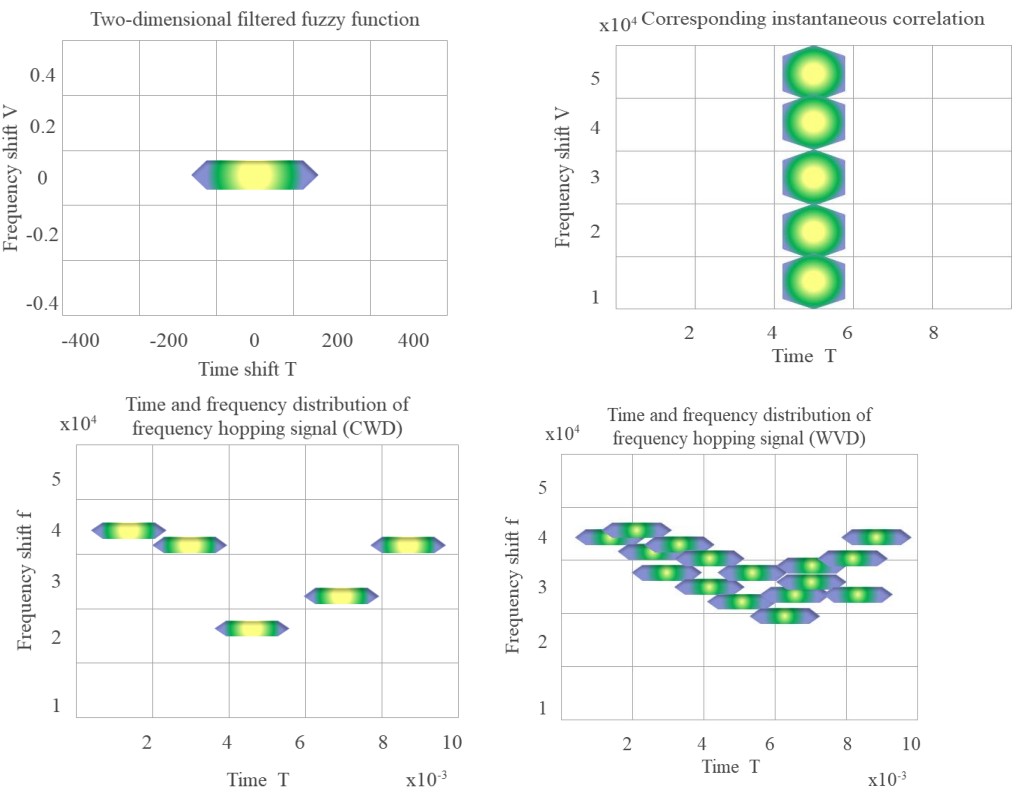

**Figure 4** **FF, instantaneous correlation function and time-frequency diagram of two distributions after two-dimensional filtering (WVD in the lower right).**

Figure 6 depicts that the larger the value of β is, the wider the central term of the KF is in the delay direction, and the smaller the value of β is, the narrower the central term of the KF is in the delay direction. Therefore, no matter how β changes, the coherent term at the far end of the delay axis cannot be suppressed. The larger the value of parameter β is, the lower the time resolution of the time-frequency distribution is, and the higher the frequency resolution is, the better the effect of suppressing the cross term is.

The parameter α controls the extension of the signal self-term along the frequency shift direction.

Since the parameter β cannot suppress the cross term at the far end of the delay axis, it is necessary to add a low-pass filter function in the delay direction, so that the KF has a better inhibitory effect on the far end of the delay axis. The improved KF is defined in Eq. (15) by using the rectangular function as the low-pass filter function.

$$\varphi(\tau,v) = \frac{sin[\alpha\pi v(\beta - |\tau|)]}{\alpha\beta\pi v} \times rect\left(\frac{\tau}{\beta}\right), \alpha \neq 0, \beta \neq 0 \tag{15}$$

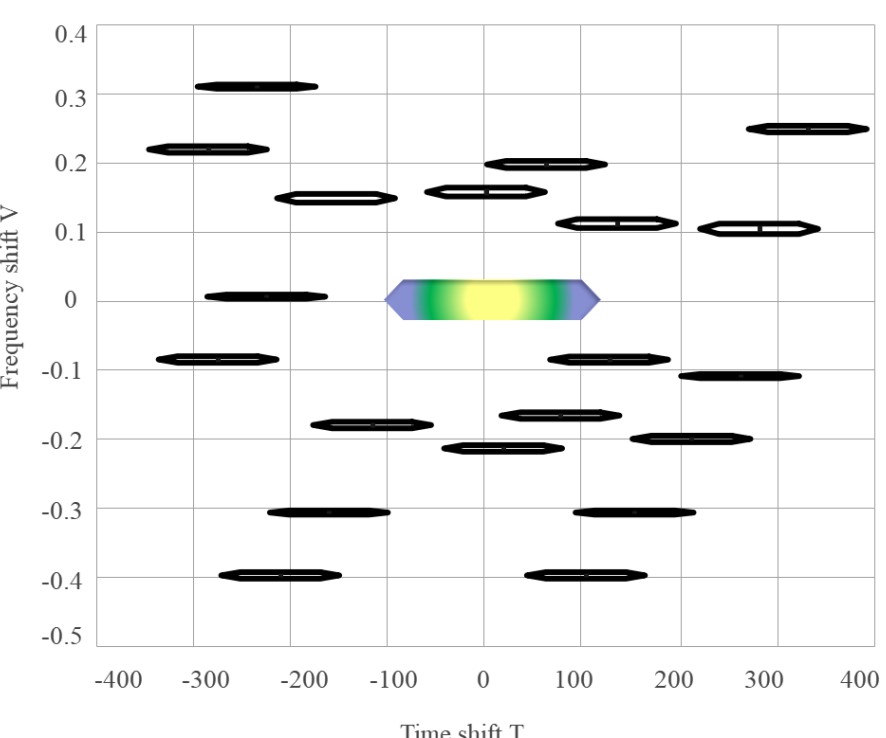

**Figure 5** FF of FHS.

The parameter β controls the width of the KF in the delay direction and the width of the rectangular function filter. The width of the low-pass filter function varies with the value of β.

## Time-frequency distribution, FF of interference, and noise signals

Noise and interfering signals have a specific time-frequency domain representation. By calculating the FFs of these signals and studying their positional relationship with the FF of the FHS and when KF is given, we can know the degree of inhibition of these signals by the KF itself.

For example, white Gaussian noise refers to noise whose probability distribution obeys the Gaussian distribution and is one of the most commonly employed mathematical models of channel noise. Figure 7 is a blur function of additive white Gaussian noise (AWGN).

Fixed-frequency signal interference means that the frequency of the interference signal is constant during the observation period, and a single fixed-frequency signal can be regarded as a stationary signal. The mathematical forms of the time domain and frequency domain are represented by Eq. (16).

$$
\begin{aligned}
&Inter_{fixed-freq}(t) = e^{j\omega_0 t}\\
&INTER_{fixed-freq}(\omega) = 2\pi\delta(\omega - \omega_0)
\end{aligned}
\tag{16}
$$

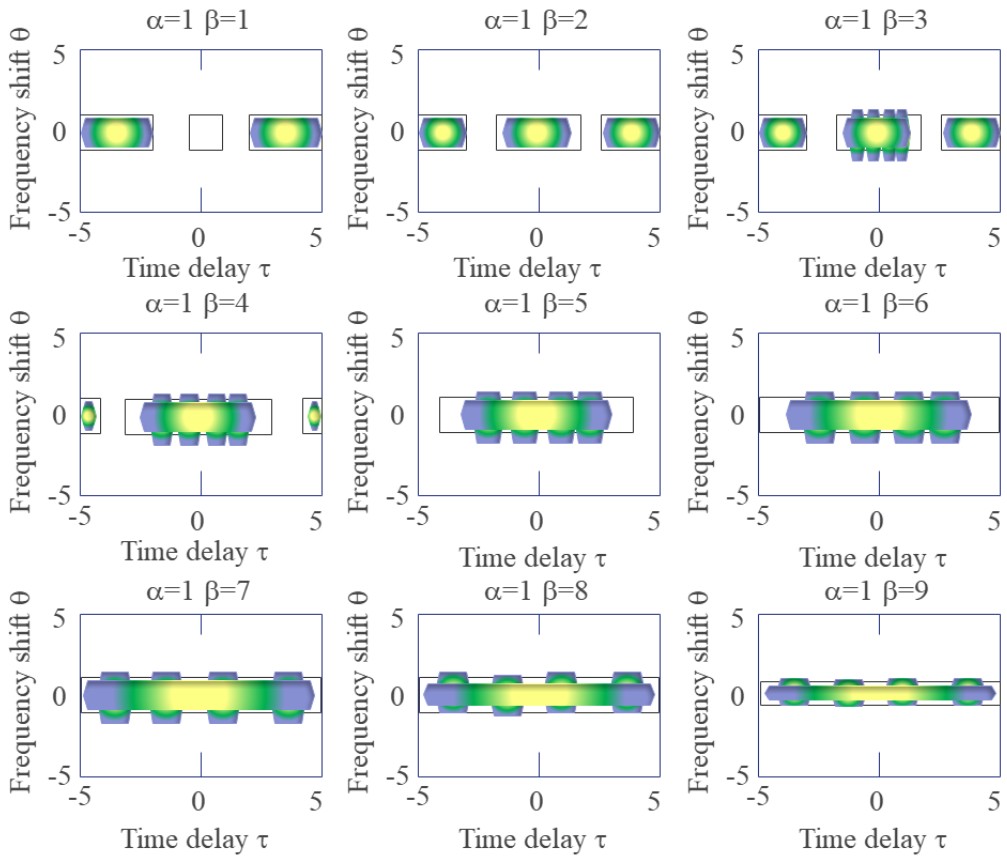

**Figure 6** Shows the effect of the parameter β on the form of the KF.

By calculating its FF, Eq. (17) is attained.

$$A_I(\tau, v) = e^{j\omega_0\tau}\delta(v) \tag{17}$$

Equations (16) and (17) depict the instantaneous frequency of the signal $\omega_0$. The simulation uses a constant-amplitude fixed-frequency interference signal whose observation duration is set to 10 ms, sampling rate is set to 200 kHz, and frequency is a constant amplitude, fixed frequency interference signal 20 kHz. Figure 8 depicts a time-frequency distribution diagram of a fixed-frequency interference signal.

Since the simulated signal is a finite-length signal, the FF of the fixed-frequency signal is an isosceles triangle when viewed from the time-shift direction, and the impulse function is viewed from the frequency-shift direction. The degree of coincidence with the KF is high near the center of the FD. The smaller the KF parameter β, the smaller the overlapping part, and the higher the suppression of fixed-frequency interference.

The frequency sweep signal interference refers to the linear change of the frequency of the interference signal during the observation period, and the mathematical form of the time domain is defined in Eq. (18).

$$Inter_{sweep-freq}(t) = e^{j\frac{1}{2}mt^2} \tag{18}$$

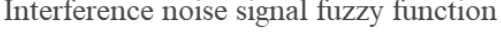

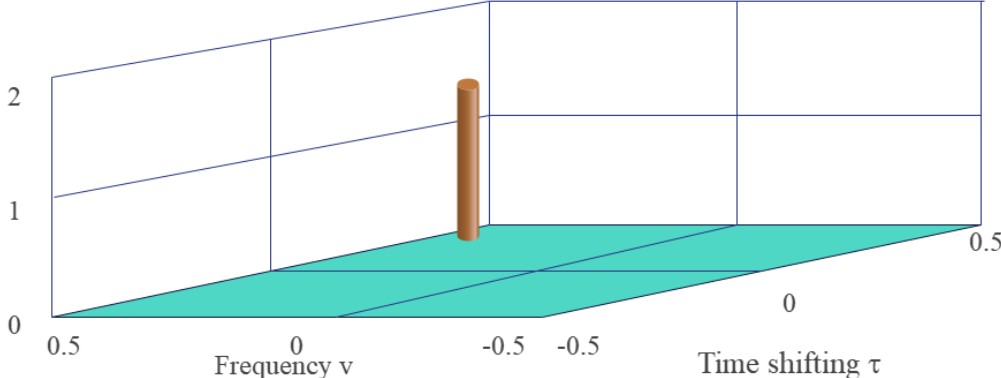

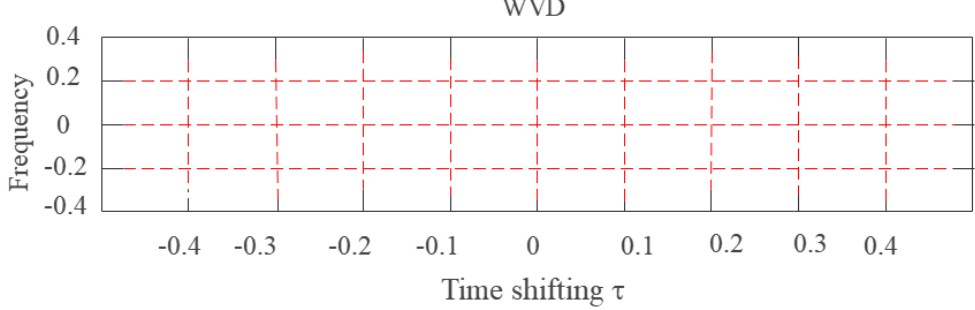

**Figure 7** FF of additive white Gaussian noise (side view and contour map).

By calculating its FF, Eq. (19) is attained.

$$A_I(\tau, v) = \delta(v - m\tau) \tag{19}$$

Equation (19) shows that the instantaneous frequency of the signal is mt. The FF is in the form of a constant-amplitude impulse function that changes linearly through the origin. The simulation uses a constant-amplitude swept-frequency interference signal whose observation duration is 10 ms, sampling rate is 200 kHz, and frequency linearly increases from initial 7 kHz to 22 kHz. Figure 9 depicts the time-frequency distribution diagram of the sweep-frequency interference signal.

Since the simulated signal represents a finite-length signal, the FF of the swept-frequency signal is an isosceles triangle when viewed from the time-shift direction, and is also an isosceles triangle when viewed from the frequency-shift direction. The degree of coincidence with the KF is low near the center of the FD, the smaller the KF parameter β, the larger the α, the smaller the overlapping part, and the higher the suppression of frequency sweep interference.

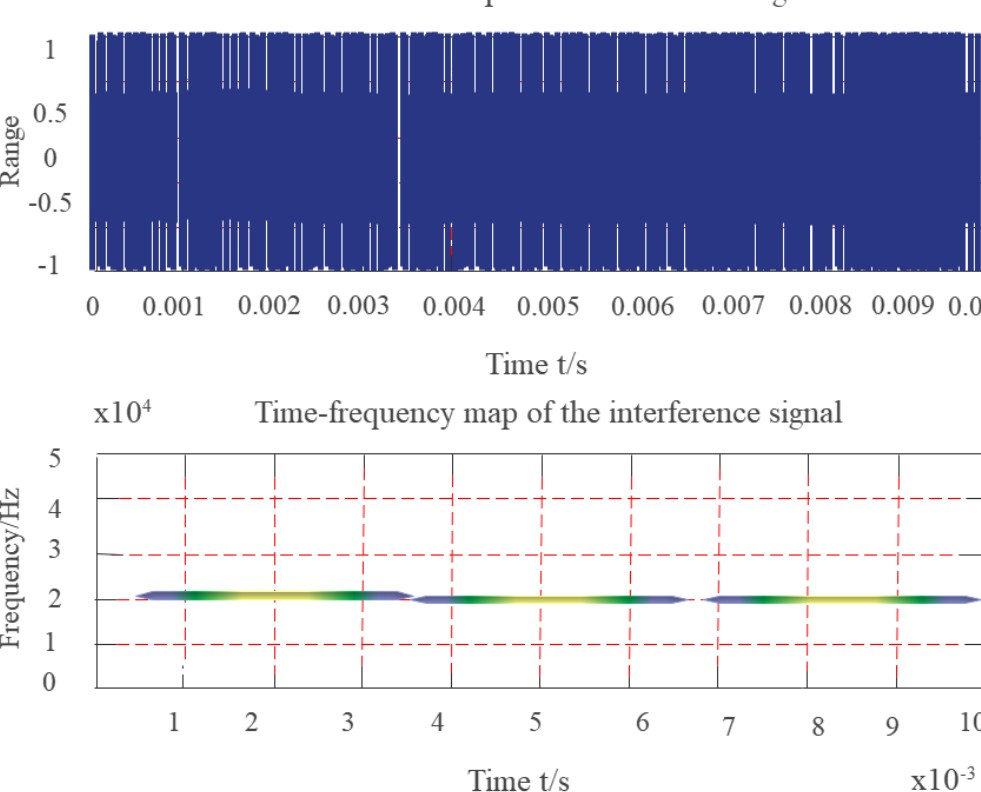

**Figure 8  Time-frequency diagram of fixed frequency signal.**

Burst interference means that the frequency of the interference signal suddenly increases in amplitude at a certain moment during the observation period. The mathematical representations of the time domain and frequency domain are defined in Eqs. (20) and (21).

$$
\begin{aligned}
Inter_{burst}(t) &= \delta(t - t_0) \\
INTER_{burst}(\omega) &= e^{-j\omega t_0}
\end{aligned}
\tag{20}
$$

By calculating its FF, Eq. (21) is attained.

$$
A_I(\tau, v) = e^{-jvt_0}\delta(\tau).
\tag{21}
$$

The simulation employs a signal whose observation duration is 10 ms, the sampling rate is 200 kHz, and burst interference occurs randomly at a certain moment.

Since the simulated signal represents a finite-length signal, the FF of the burst signal is an impulse function when viewed from the time-shift direction, and an isosceles triangle when viewed from the frequency-shift direction. The degree of coincidence with the KF near the center of the FD is low, the larger the KF parameter $\alpha$, the smaller the overlapping part, and the higher the suppression of burst interference.

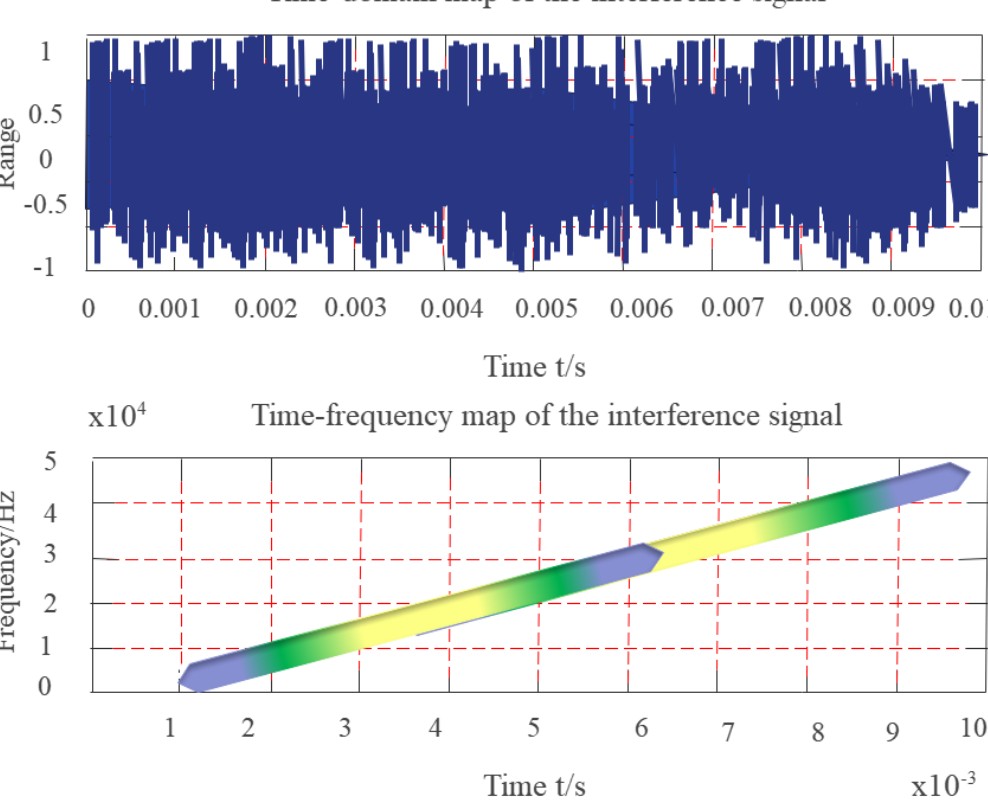

**Figure 9** **Time-frequency diagram of swept frequency signal.**

The non-linear frequency sweep signal interference refers to the non-linear frequency change of the interference signal during the observation period. The mathematical representation of the time domain is defined in Eq. (22).

$$Inter_{sweep-freq}(t) = e^{j\frac{1}{3}mt3} \tag{22}$$

Equation (22) shows that the instantaneous frequency of the signal is mt2, and its FF is in the form of a nonlinear change impulse function through the origin. The simulation employs a constant-amplitude nonlinear swept-frequency interference signal with an observation duration of 10 ms, a sampling rate of 200 kHz, and a non-linear increase in frequency from an initial 1.5 kHz to 25 kHz. Figure 10 depicts a time-frequency diagram of a nonlinear sweep-frequency signal.

Since the simulated signal represents a finite-length signal, the maximum value of the FF of the nonlinear frequency swept signal is located at the origin of the FD, and the FF is symmetrical about the origin. The degree of coincidence with the KF near the center of the FD is low, the smaller the KF parameter β, the larger the α, the smaller the overlapping part, and the higher the suppression of nonlinear frequency sweep interference.

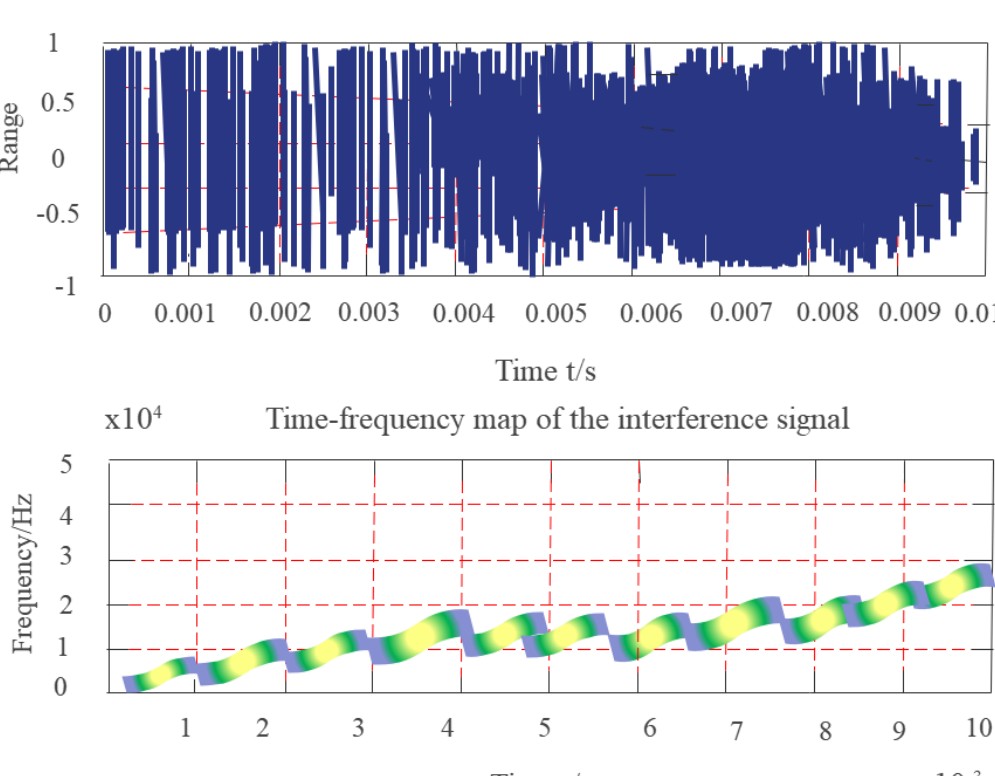

**Figure 10 Time-frequency diagram of nonlinear frequency sweep signal.**

## EXPERIMENTAL RESEARCH

According to the research configuration, it is necessary to analyze and verify the spectral distribution and pulse compression characteristics, Doppler tolerance, and cross-correlation characteristics of random sequence chirp signals. Based on the above requirements, the following simulation platform is built. In the system, an arbitrary waveform generator is implemented to generate the intermediate frequency waveform. After up-conversion, the signal is connected to the down-conversion module by employing antenna radiation or direct injection, and then the oscilloscope is employed for sampling, finally, the sampling data is analyzed by the computer. If the signals interfere with each other seriously, the research purpose will be lost. We employ the two channels of the arbitrary waveform generator to transmit two signals with different random sequences, but the pulse envelopes are superimposed on each other, and the cross-correlation performance

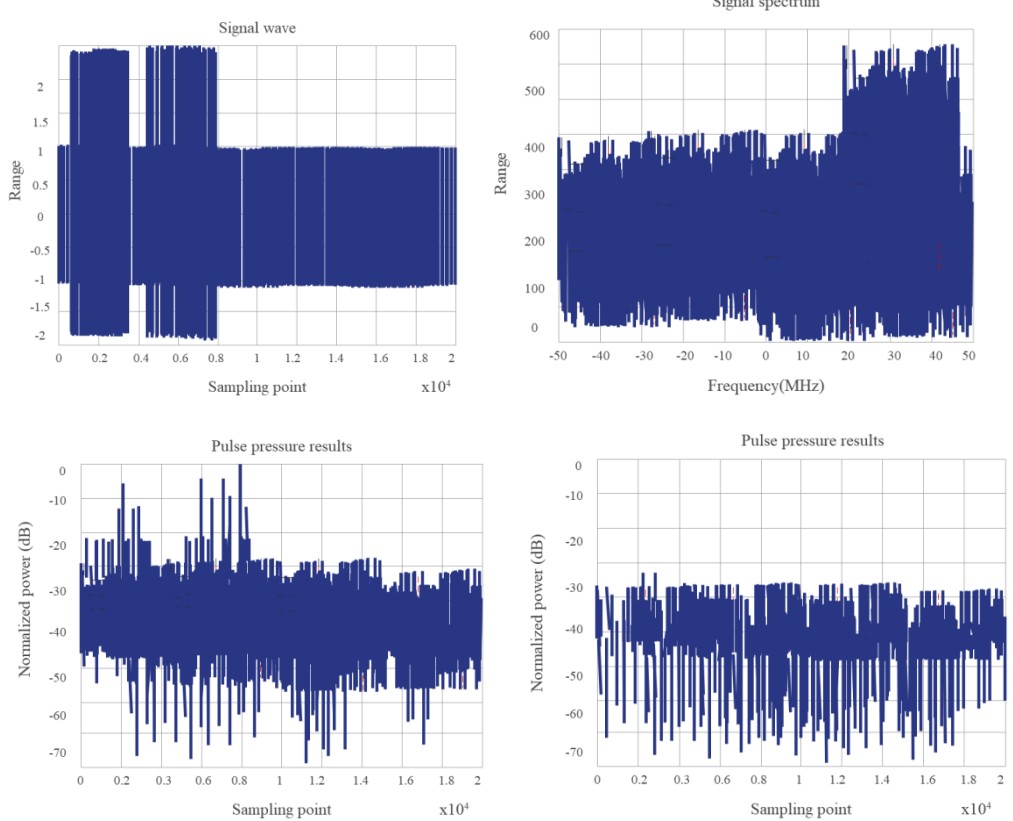

**Figure 11 The envelopes of the two pulse signals do not overlap.**

is verified by frequency conversion sampling processing. Figures 11 and 12 show the test results of the system scheme.

It is verified that the combinatorial mathematical model for computer random signals shows a good effect and can effectively improve the effect of random signal combinations (*Roy, Kumar & Chang, 2020*).

## CONCLUSION

From the perspective of digital signal processing, the manuscript proposes a modeling tool called a combinatorial mathematical algorithm for computer random signals. The conventional combinatorial mathematical model can run the basic analysis, but the error is relatively large and the precision is not high, so it is necessary to examine the combinatorial mathematical model to control computer random signals. By proposing the mathematical combination algorithm the time characteristic function of the computer random signal can be transformed into a novel form of linear expression. The article shows the computational validity of the control-based combinatorial mathematical model for random computer signals by running experiments and verifying the convergence speed and control efficiency.

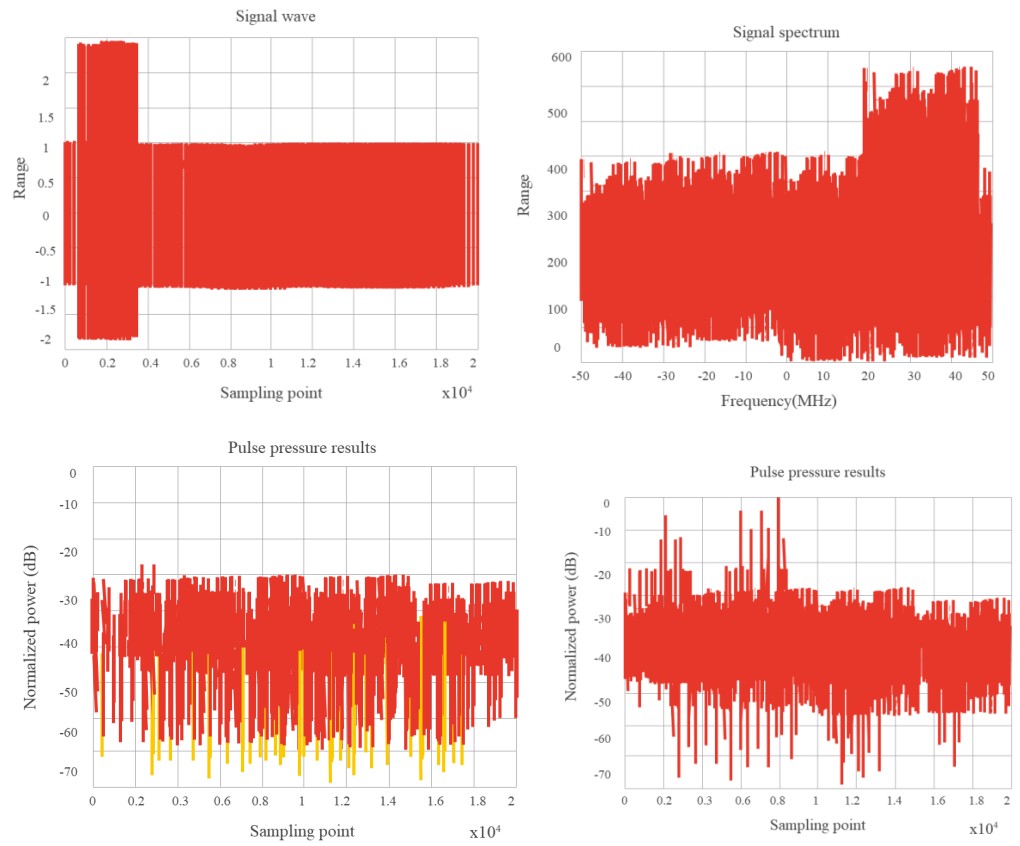

**Figure 12  95% overlap of two pulse signal envelopes.**

Furthermore, through experimental research, the combinatorial mathematical model for computer random signals can effectively combine random signals.

### Funding

The study was supported by the Department of Education of Henan Province (No. 2021SJGLX869). The funders had no role in study design, data collection and analysis, decision to publish, or preparation of the manuscript.

### Grant Disclosures

The following grant information was disclosed by the authors:
Department of Education of Henan Province: 2021SJGLX869.

### Competing Interests

The authors declare there are no competing interests.

## Author Contributions

- Qinghua Yao conceived and designed the experiments, performed the experiments, analyzed the data, performed the computation work, prepared figures and/or tables, authored or reviewed drafts of the article, and approved the final draft.
- Benhua Qiu conceived and designed the experiments, performed the experiments, analyzed the data, performed the computation work, prepared figures and/or tables, authored or reviewed drafts of the article, and approved the final draft.

## Data Availability

The code and dataset are available in the Supplemental Files.

## Supplemental Information

Supplemental information for this article can be found online at http://dx.doi.org/10.7717/peerj-cs.1873#supplemental-information.

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
