# Peer review of "Algorithm design of a combinatorial mathematical model for computer random signals"

_PeerJ Computer Science, doi:10.7717/peerj-cs.1873_

## Round 0.1 · original submission · Major Revisions

Based on the reviewers’ comments, you may resubmit the revised manuscript for further consideration. Please consider the reviewers’ comments carefully and submit a list of responses to the comments along with the revised manuscript.

**Language Note:** The review process has identified that the English language must be improved. PeerJ can provide language editing services - please contact us at [email protected] for pricing (be sure to provide your manuscript number and title). Alternatively, you should make your own arrangements to improve the language quality and provide details in your response letter. – PeerJ Staff

Reviewer 1 ·

Basic reporting

The paper addresses the enhancement of computer random signal processing by integrating an intelligent signal recognition algorithm into the design of a combinatorial mathematical model. Specifically, the focus is on studying the parameter estimation of conventional Frequency Hopping Signals (FHS) through the utilization of an optimization kernel function (KF). The research begins by exploring the mathematical form and graphical representation of the ambiguity function associated with conventional FHS.

Experimental design

The experimental findings demonstrate that the combinatorial mathematical model algorithm for computer random signal presented in this paper yields positive results. The approach effectively improves the processing of random signals, particularly in the context of various interference noise scenarios, however following improvements will be beneficial

1. How did the authors validate that the signal’s distribution is a specific distribution called “Choi-Williams”. Please discuss it. Did the Authors run any method to test it?
2. What does “Using MATLAB language simulation (α=0.5)” mean? What is the alpha parameter used for? Please discuss it.
3. All figures should be checked and fixed. Some words in the graphs are misspelled.
4. How did the authors fuzzify the functions? Please discuss it in the text.
5. Why did the authors use both time and frequency domains together since all analyses are generally based on one of them by changing time to frequency or frequency to time? What advantages are derived from utilizing both? Please discuss more.
6. Why did the authors use fuzzy set theory in functions? What advantages are derived from? Please discuss more.
7. Did the authors fuzzify the function domain and fuzzy range? Please provide more remarks
8. What is a “combinatorial mathematical model”? Even though it is widely underlined in the text, there is no such exact definition or formulation found in the article. Please discuss more.

Validity of the findings

The validity seems fine

Additional comments

1. The abstract should summarize what the research content is about mentioning the research motivation, research problem, proposed method, data set, contribution, and key findings. The abstract should be rewritten and reorganized.
2. The first paragraph of the introduction section should be removed.
3. The introduction section should contain two more paragraphs: 1. The research motivation and the contribution, 2. How the rest of the article is outlined. The last paragraph of the introduction should be removed.
4. All equations should be cited in the text. Abbreviations should not be used in the titles of sections and subsections. All words should be presented fully before abbreviations. The abbreviation, Eq. (.) should be used in the text instead of the word “formula”.
5. All titles of sections and subsections should be checked and fixed.
6. Proofreading is a must.

Reviewer 2 ·

Basic reporting

Please provide detailed responses to issues that are itemized as follows:
1. This sentence is taken from the text: “The Gaussian in White Gaussian Noise refers to the noise whose probability distribution obeys the Gaussian normal distribution” This is not correct. There are several sentences in the text that are not correct scientifically. All should be checked and fixed.
2. Authors claimed that “simulation experiments are carried out in different types of interference noise environments”. What are they? Please provide more remarks and discussions. How did the proposed method react to those interference noise environments? Please discuss more.
3. The introduction section has redundant information and remarks. They need to be removed. The introduction section should be rewritten and reorganized. More up-to-date references should be added and discussed. Moreover, the literature review should be conducted in an informative manner not listing studies in an order.
4. The authors presented Eq. (1) using this sentence “The definition of the FF is:” This is not a definition, but it is a mathematical expression. Those sentences are misleading the readers. All should be checked and fixed.
5. Even though there are terms used fuzzy in the figures, how the authors use fuzzy and fuzzification is not clear. Please discuss more remarks or remove them.
6. What is the “instantaneous correlation function”? how is it defined? Please provide more information.

Experimental design

7. What are the parameters of Wigner-Ville and Choi-Williams distributions? Are they continuous or discrete? Please discuss them.
8. This sentence is taken from the text:” Substituting the formula into formula (1),” Which formula? All should be checked and fixed.
9. All mathematical expressions should be clearly presented, so their reliability of them increases.

Validity of the findings

10. Authors claimed that “The parameter α controls the extension of the signal self-term along the frequency shift direction.” and “The parameter β controls the width of the KF in the delay direction and the width of the rectangular function filter. The width of the low-pass filter function varies with the value of β.” Did they run any analyses and tabulate the results? If not, why did not run the analysis since it improves the conducted research results?
11. The article needs a complete revision in terms of language, presentation, and template.
12. Which type of fuzzy membership function was used in the analysis? How did the authors decide to use it, empirically or theoretically?

---

## Round 0.2 · accepted · Accept

The reviewers are satisfied with the revisions and have recommended the accept decision, Congratulations.

Reviewer 1 ·

Basic reporting

The authors have revised the manuscript according to the previous inputs.

Experimental design

Experimentation is done well revised.

Validity of the findings

The authors have conducted a thorough and well-organized study that significantly contributes to the existing literature in computer science. The research methodology is sound, and the results are presented with clarity. The findings of the study are both valuable and relevant to the field.

Additional comments

I appreciate the authors' attention to detail, the robustness of their analysis, and the overall quality of the manuscript. I believe that this article will make a meaningful contribution to our journal and will be of interest to readership.

Reviewer 2 ·

Basic reporting

Thank you for making efforts in addressing the raised comments.

Experimental design

no comments

Validity of the findings

no comments